# SiR-DNA/SiR–Hoechst-induced chromosome entanglement generates severe anaphase bridges and DNA damage

Girish Rajendraprasad[1],*, Sergi Rodriguez-Calado[1],*, Marin Barisic[1,2]

**SiR-DNA/SiR–Hoechst is a far-red fluorescent DNA probe that is routinely used for live-cell imaging of cell nuclei in interphase and chromosomes during mitosis. Despite being reported to induce DNA damage, SiR-DNA has been used in more than 300 research articles, covering topics like mitosis, chromatin biology, cancer research, cytoskeletal research, and DNA damage response. Here, we used live-cell imaging to perform a comprehensive analysis of the effects of SiR-DNA on mitosis of four human cell lines (RPE-1, DLD-1, HeLa, and U2OS). We report a dose-, time-, and light-dependent effect of SiR-DNA on chromosome segregation. We found that, upon the exposure to light during imaging, nanomolar concentrations of SiR-DNA induce non-centromeric chromosome entanglement that severely impairs sister chromatid segregation and spindle elongation during anaphase. This causes DNA damage that is passed forward to the following cell cycle, thereby having a detrimental effect on genome integrity. Our findings highlight the drawbacks in using SiR-DNA for investigation of late mitotic events and DNA damage-related topics and urge the use of alternative labeling strategies to study these processes.**

## Introduction

Live-cell imaging enables visualization of dynamic cellular processes in real time and has therefore been extensively used by cell biologists. The development of live-cell–compatible fluorescent probes for DNA, such as SiR–Hoechst (also known as SiR-DNA), has streamlined the imaging of nuclei/chromosomes, avoiding often complex and time-consuming generation of stable cell lines or transient transfections used to (over)express fluorescent proteins ([1]). Far red light-excited SiR-DNA is a cell-permeable DNA probe that has been widely adopted for live-cell imaging due to its

advantages over earlier generation of Hoechst-based dyes. Compared with, for example, Hoechst 33342, SiR-DNA is compatible with live-cell super-resolution microscopy and displays minimal cytotoxicity by avoiding DNA damage induced by UV light ([1], [2], [3], [4], [5]). Thus far, SiR-DNA has been used in more than 300 research articles (Royal Danish Library database) covering topics such as mitosis, chromatin biology, cancer research, cytoskeletal research, and DNA damage response. Although initial studies suggested that SiR-DNA is noncytotoxic and has minimal influence on mitotic duration at concentrations as high as 10 μM, a recent study showed that SiR-DNA can induce DNA damage and impair cell cycle progression, raising concerns about the safety and validity of its use in live-cell imaging experiments ([6]). These findings highlighted the need for an extensive revisiting of potential risks and limitations of SiR-DNA usage and evaluation of its effects on sensitive cellular processes, such as mitosis.

In this study, we performed a detailed analysis of the effects of SiR-DNA on mitosis using spinning-disk confocal live-cell imaging in four human cell lines that are commonly used in research (RPE-1, DLD-1, HeLa, and U2OS). We observed a dose-, time-, and light-dependent effect of SiR-DNA on chromosome segregation that led to increased levels of DNA damage. Specifically, we found that, upon the exposure to light during imaging, nanomolar concentrations of SiR-DNA induced non-centromeric chromosome entanglement that severely impaired sister chromatid segregation and spindle elongation during anaphase, resulting in DNA damage that was passed forward to the following cell cycle, thereby having a detrimental effect on genome integrity.

## Results

### SiR-DNA induces severe chromatin bridges (SCBs) during mitosis

To monitor the impact of SiR-DNA staining on dividing cells, we performed spinning-disk confocal live-cell imaging of mitosis in

[1]Cell Division and Cytoskeleton, Danish Cancer Institute, Copenhagen, Denmark [2]Department of Cellular and Molecular Medicine, Faculty of Health and Medical Sciences, University of Copenhagen, Copenhagen, Denmark

Correspondence: girish@cancer.dk; barisic@cancer.dk
*Girish Rajendraprasad and Sergi Rodriguez-Calado contributed equally to this work

four frequently used human cell lines (RPE-1, DLD-1, U2OS, and HeLa) treated with different concentrations of SiR-DNA for varying incubation time. We analyzed chromosome movement, mitotic duration, and cell fate in SiR-DNA–labeled cells, compared with cells with adenovirus-based expression of histone 2B (H2B)-RFP as controls. Both SiR-DNA-labeled cells and cells expressing H2B-RFP displayed normal behavior in early mitosis, forming a proper bi-polar spindle and congressing the chromosomes to metaphase plate within the expected time frame. However, whereas H2B-RFP–expressing cells undertook normal metaphase-to-anaphase transition and were successfully divided, all four studied SiR-DNA-labeled cell lines displayed SCBs in anaphase, which obstructed chromosome segregation and cytokinesis. These SCBs were characterized by the presence of a large amount of chromatin apparently connecting all segregating chromatids (Fig 1A and B, Video 1, Video 2, Video 3, and Video 4) and were thus different from the chromatin bridges typically occurring in cancer cell lines (Fig S1A). Importantly, the effect of SiR-DNA on chromosome segregation was dose- and time-dependent (Figs 1A and B and S1A). Whereas cells treated with 20 nM SiR-DNA for 2 h displayed some prevalence of SCBs (16.04% ± 9.90% in RPE-1; 16.87% ± 2.76% in U2OS; 21.28% ± 12.12% in DLD-1; 15.11% ± 4.41% in HeLa), there was a substantial increase in the incidence of SCBs upon treatment with higher concentrations (Fig 1B). Cells labelled with 100 nM SiR-DNA for 30 min already showed higher rates of SCBs (36.31% ± 28.93% in RPE-1; 34.47% ± 8.38% in U2OS; 20.00% ± 10.00% in DLD-1; 67.10% ± 5.78% in

HeLa) and increasing the duration of incubation to 2 h displayed comparable levels of SCB occurrence (58.89% ± 18.36% in RPE-1; 45.38% ± 5.04% in U2OS; 41.98% ± 10.63% in DLD-1; 72.48% ± 3.29% in HeLa) (Fig 1B). 500 nM SiR-DNA-labeled cells showed significant increase in SCBs both upon 30-min (57.94% ± 8.36% in RPE-1; 87.88% ± 10.50% in U2OS; 54.55% ± 7.88% in DLD-1; 83.05% ± 5.99% in HeLa) and 2 h incubations (73.28% ± 13.57% in RPE-1; 88.43% ± 11.14% in U2OS; 80.08% ± 3.21% in DLD-1; 77.03% ± 10.42% in HeLa) in all tested cell lines (Fig 1B). Noteworthily, a drop in SCB incidence was observed upon the overnight treatment with SiR-DNA, an effect that could be explained by the dye being pumped-out, as this was coupled to a decrease in the signal-to-noise ratio of nuclear stain (Fig S1B). The observed negative effects of SiR-DNA were limited to anaphase defects, as the overall duration of mitosis remained unperturbed among the tested conditions in all studied cell lines (Fig S1C), thus being in line with a previous similar observation (6).

Taken together, these results revealed strong side effects of SiR-DNA-based chromosome labeling, which can interfere with chromosome segregation by inducing chromosome entanglement that leads to SCBs.

## SiR-DNA–derived SCBs are induced by light irradiation

Next, we tested whether the SiR-DNA–induced SCBs depend on light irradiation during imaging. To monitor mitosis in more detail, we

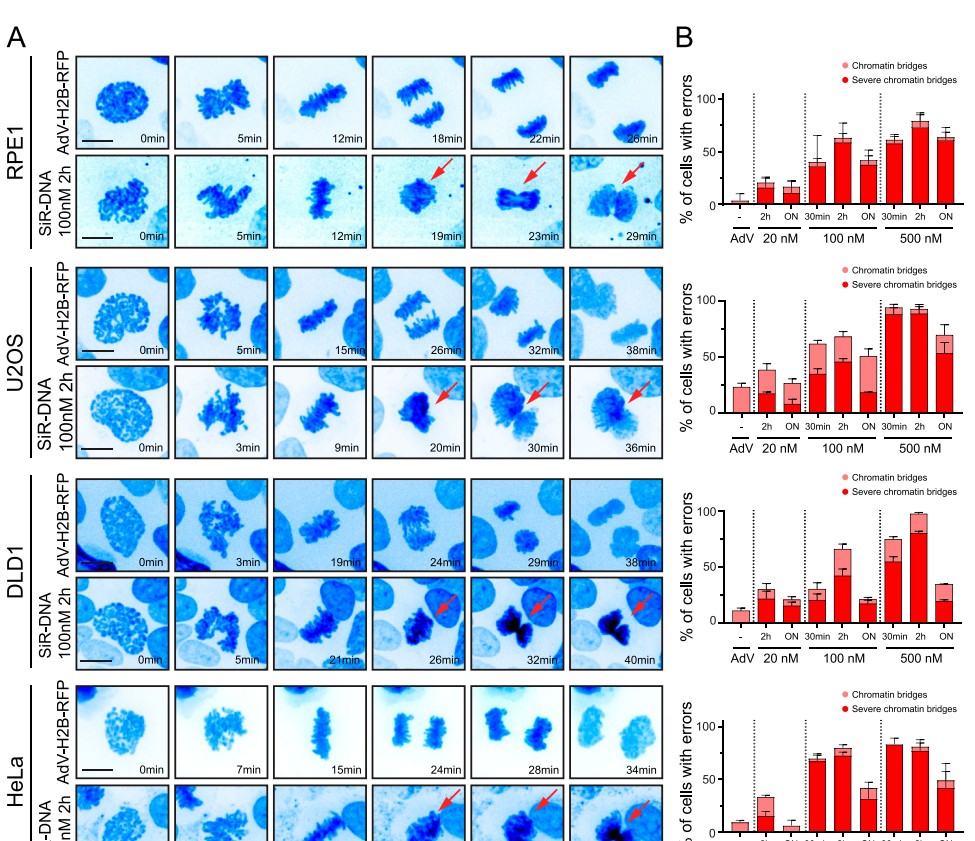

**Figure 1. SiR-DNA induces severe chromatin bridges (SCBs) in a time- and dose-dependent manner.**

**(A)** Representative spinning-disk confocal time-series of mitosis in indicated cell lines either infected with adenovirus to express H2B-RFP or incubated with 100 nM SiR-DNA for 2 h. To achieve proper staining in U2OS and DLD1 cells, efflux pumps were inhibited using verapamil. Arrows highlight the presence of SCBs. Scale bars 10 μm. **(B)** Quantification of chromatin bridges and SCBs upon SiR-DNA treatment with indicated concentrations and durations. Bar graphs with SD are plotted from three independent experiments. ON, overnight.

used RPE-1 and U2OS cells stably expressing fluorescently tagged H2B and α-tubulin. RPE-1 H2B-GFP/RFP-α-tubulin and U2OS H2B-GFP/mCherry-α-tubulin cells were incubated with 100 nM SiR-DNA for 2 h and imaged with or without 640 nm laser light irradiation. As expected, during the 640 nm laser-based live-cell imaging, we observed SCBs in both SiR-DNA–treated cell lines (48.15% ± 3.21% in RPE-1 H2B-GFP/RFP-α-tubulin; 66.07% ± 4.26% in U2OS H2B-GFP/mCherry-α-tubulin) (Fig 2A and B). However, similar to non-treated cells irradiated with the 640 nm laser, SCBs were absent in both SiR-DNA-treated cell lines that were not imaged using the 640 nm laser (Fig 2A and B). Moreover, imaging using a wider time interval (5 min) showed a significant drop in SCB rates in both cell lines (12.34% ±

2.65% in RPE-1 H2B-GFP/RFP-α-tubulin; 14.49% ± 2.09% in U2OS H2B-GFP/mCherry-α-tubulin) (Fig 2A and B).

These results demonstrate that a mere application of the dye is not sufficient to induce SCBs, and that the observed toxic effect of SiR-DNA on chromosome segregation depends on light irradiation.

## SiR-DNA impairs anaphase spindle elongation

To further investigate the impact of SiR-DNA-induced problems on anaphase, we analyzed spindle elongation, a process that drives chromosome segregation in human cells (7, 8) and contributes

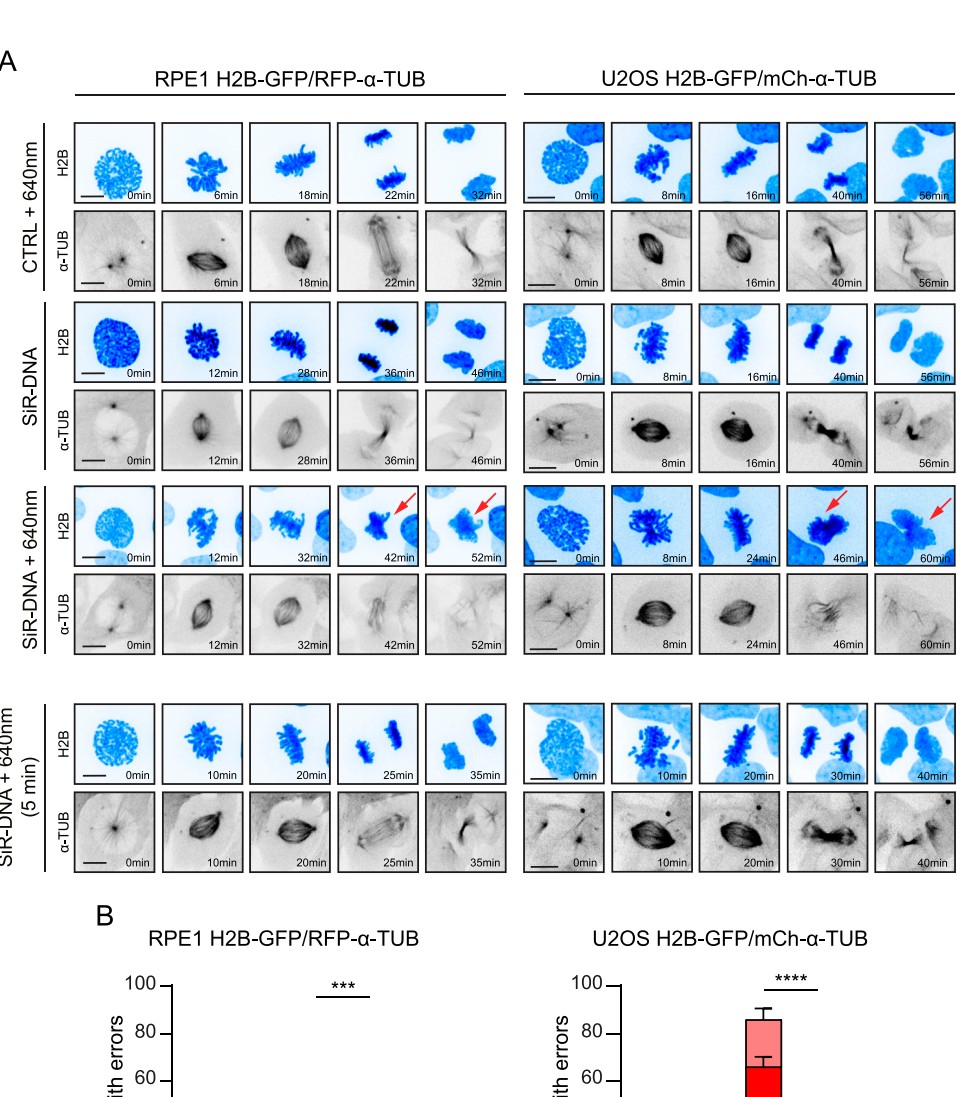

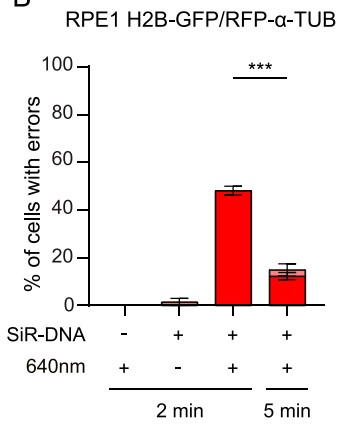

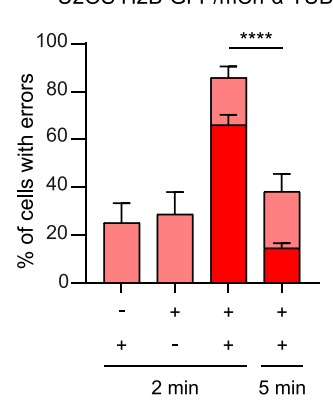

**Figure 2. Severe chromatin bridges are observed exclusively upon 640 nm light irradiation of SiR-DNA.**
**(A)** Representative spinning-disk confocal time-series of mitosis in RPE-1 H2B-GFP/RFP-α-tubulin (left) and U2OS H2B-GFP/mCherry-α-tubulin (right) in the presence or absence of 640 nm laser irradiation after incubation with 100 nM SiR-DNA for 2 h. Imaging was performed at either 2- (top) or 5-min (bottom) interval. Arrows highlight the presence of severe chromatin bridges. Scale bars 10 μm. **(B)** Quantification of chromatin bridges and severe chromatin bridges upon SiR-DNA treatment in the presence or absence of 640 nm laser light in the indicated time intervals and cell lines. Bar graphs with SD are plotted from three independent experiments. P-values were calculated using t test. ***P ≤ 0.0001, ****P < 0.0001.

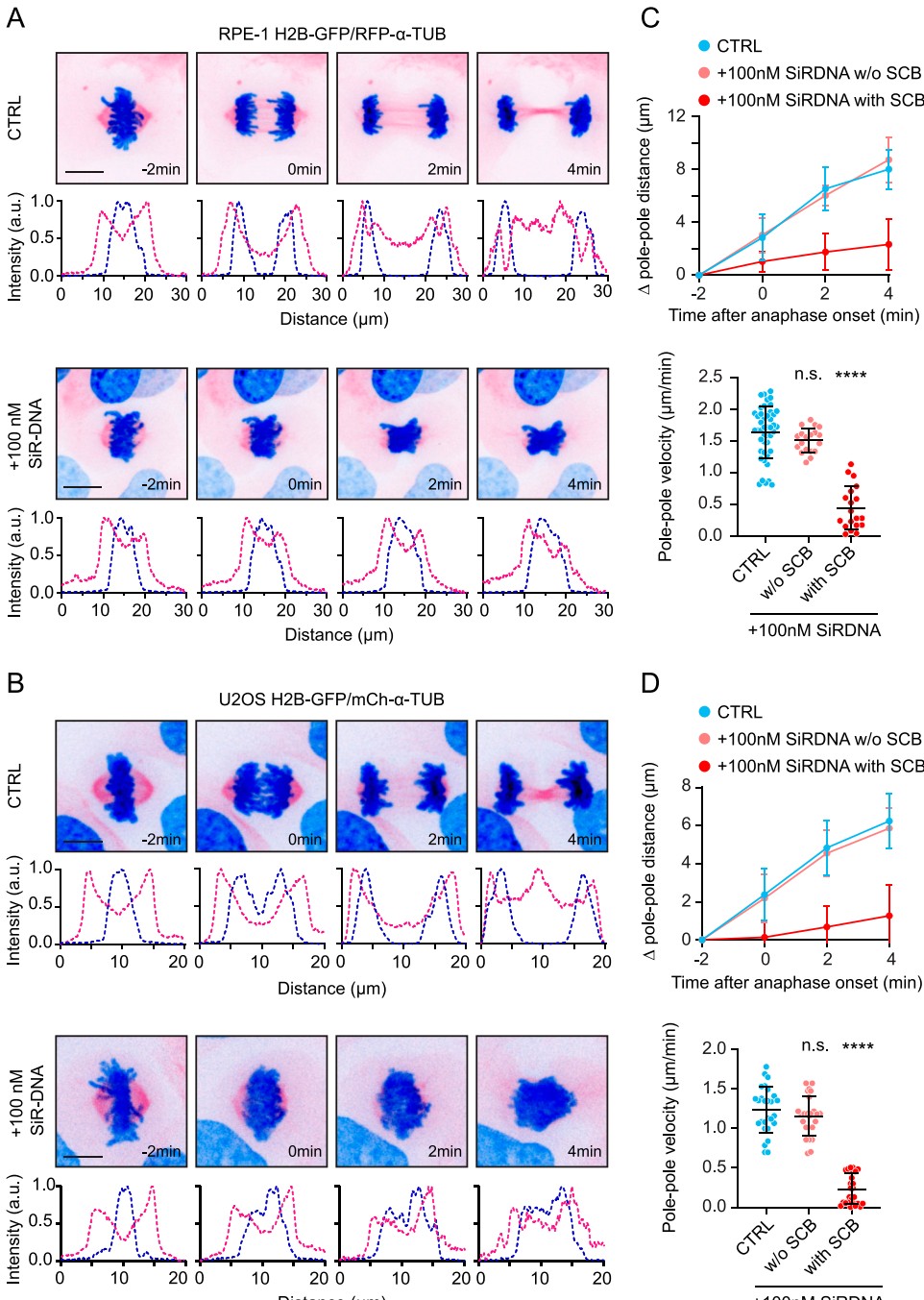

to the correction of erroneous chromosome attachments in mitosis (9). We performed live-cell imaging of RPE-1 H2B-GFP/RFP-α-tubulin and U2OS H2B-GFP/mCherry-α-tubulin cells treated with SiR-DNA and monitored spindle elongation upon the anaphase onset. By measuring the distance and velocity of spindle elongation in the presence or absence of the dye, we revealed that SiR-DNA–induced SCBs strongly impaired spindle elongation during anaphase (0.44 ± 0.34 μm/min in SiR-DNA–treated cells versus 1.64 ± 0.41 μm/min in control cells in RPE-1 H2B-GFP/RFP-α-tubulin; 0.23 ± 0.20

μm/min in SiR-DNA–treated cells versus 1.25 ± 0.29 μm/min in control cells in U2OS H2B-GFP/mCherry-α-tubulin), in both cell lines (Fig 3A–D, Video 5, and Video 6). Of note, SiR-DNA staining affected spindle elongation exclusively in cells displaying SCBs, whereas it remained unperturbed in the fraction of cells devoid of SCBs (RPE-1 H2B-GFP/RFP-α-tubulin: CTRL—1.64 ± 0.41 μm/min, w/o SCB—1.51 ± 0.19 μm/min; U2OS H2B-GFP/mCherry-α-tubulin: CTRL—1.25 ± 0.29 μm/min, w/o SCB—1.16 ± 0.28 μm/min) (Fig 3A–D, Video 5, and Video 6).

**Figure 3. Severe chromatin bridges block anaphase spindle elongation.**
**(A, B)** Representative spinning-disk confocal time-series of anaphase in control and cells with severe chromatin bridge after SiR-DNA treatment. Corresponding line intensity profile of tubulin (magenta) and DNA (blue) across the spindle is shown below. Scale bars 10 μm. **(C, D)** Quantification of the anaphase spindle elongation distance over time and velocity in the indicated cell lines. *P*-values were calculated using Mann–Whitney *U* test. n.s., not significant, ****P* < 0.0001.

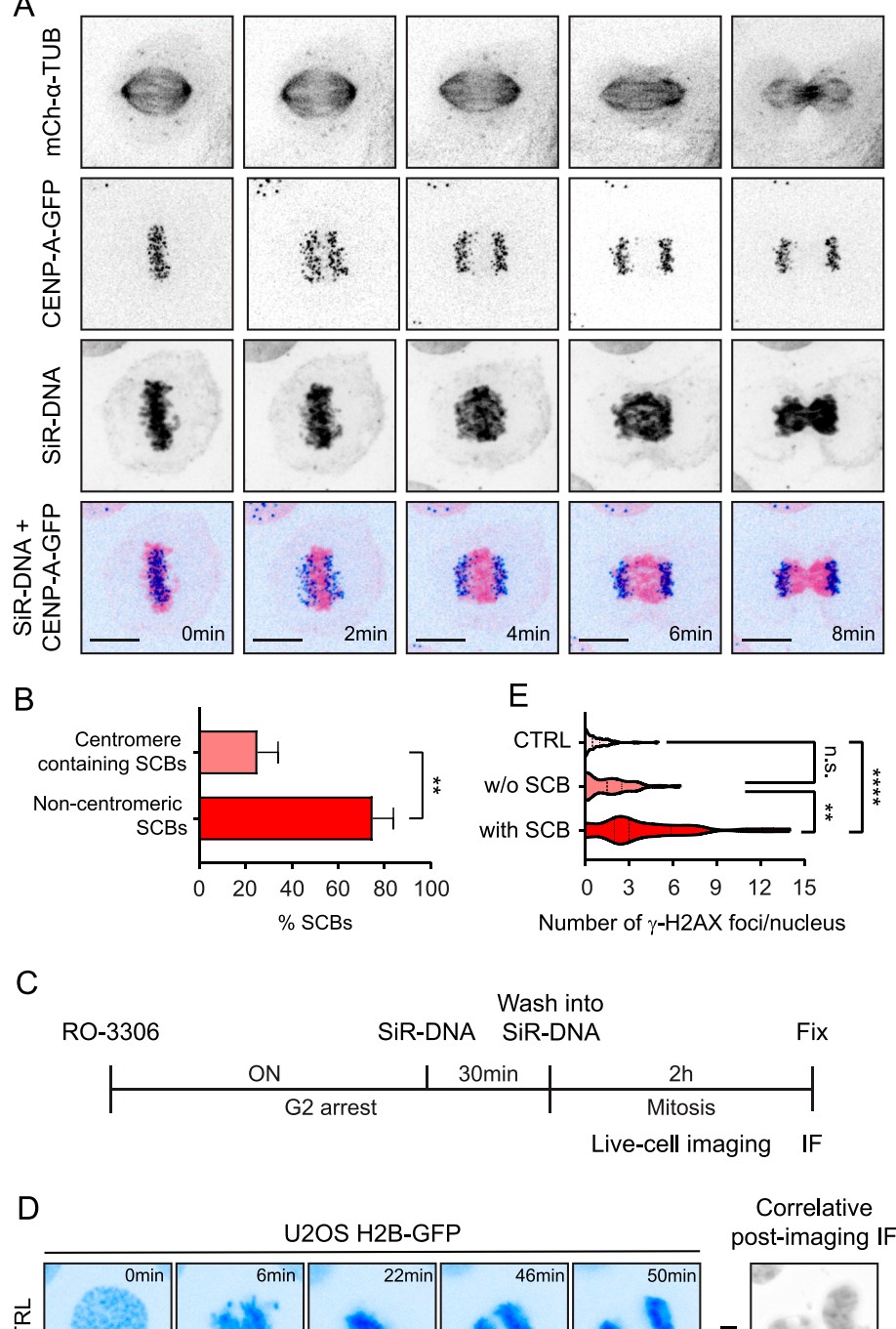

**Figure 4. SiR-DNA–derived severe chromatin bridges are caused by non-centromeric chromosome entanglement and induce DNA damage during cell division.**
**(A)** Time-lapse series of anaphase in U2OS CENP-A-GFP/mCherry-α-tubulin cells treated with 100 nM SiR-DNA for 2 h. Scale bar 10 μm. **(B)** Quantification of percentage of severe chromatin bridges with centromeres marked by CENP-A found in central spindle in anaphase. P-values were calculated using t test. **P ≤ 0.001. **(C)** Schematic illustration for correlative microscopy protocol. IF, immunofluorescence; ON, overnight. **(D)** Time-lapse series of control and SiR-DNA–treated U2OS H2B-GFP cells (left panel). After the live-cell imaging, cells were fixed and correlative immunofluorescence of DNA damage marker γ-H2AX was performed (right panel). Scale bars 10 μm. **(E)** Quantification of the number of γ-H2AX foci per nucleus upon SiR-DNA treatment. n.s., not significant, ****P < 0.0001.

These results indicate that SiR-DNA–induced chromosome entanglement impairs chromosome segregation and, consequently, prevents anaphase spindle elongation.

### SiR-DNA induces non-centromeric chromosome entanglement that leads to DNA damage

Next, we aimed to elucidate the consequences of SiR-DNA–induced SCBs during chromosome segregation. The observed chromosome segregation problems may originate at centromeric regions or may arise from the entanglements of other chromatin regions, such as telomere fusions (10, 11). Chromosome bridge formation and its eventual breakage can result in DNA damage because of mechanical fragmentation and aberrant DNA replication at the broken sites (12). To assess the nature of the observed chromosome entanglement induced by SiR-DNA, we performed live-cell imaging of U2OS cells stably expressing CENP-A-GFP, a centromere marker which enables us to monitor the impact of the dye on centromere separation during anaphase. In most of the cells, all centromeres were able to correctly separate in anaphase, regardless of the formation of SCBs (74.92% ± 19.95% of U2OS CENP-A-GFP cells presenting SCB), whereas a smaller proportion of SCB-containing cells displayed few centromeres remaining in the central spindle region after the anaphase onset (Figs 4A and B and S2). These results indicate that SiR-DNA mainly induces non-centromeric chromosome entanglement (Fig 4A and B).

To analyze the effect of SiR-DNA–induced chromosome entanglement on genomic integrity, we performed live-cell imaging followed by correlative imaging of a DNA damage marker γ-H2AX (H2AX pS139). To exclude any potential effect of SiR-DNA on the DNA damage arising before mitosis (e.g., during DNA replication), we exposed U2OS H2B-GFP cells to SiR-DNA at the G2/M phase border using CDK1 inhibitor RO-3306 (Fig 4C). After the live-cell imaging to monitor the mitotic process, cells were fixed and stained for γ-H2AX to detect DNA damage. Whereas control cells presented low levels of DNA damage after cell division, the amount of γ-H2AX-positive foci highly increased in cells with SiR-DNA–induced SCBs (4.0 ± 3.5 foci/nucleus compared with 0.7 ± 1.0 foci/nucleus in control cells) (Fig 4D and E). Of note, increased DNA damage was a direct consequence of SiR-DNA–induced SCBs, as SiR-DNA–treated cells that divided normally, displaying no SCBs, did not show statistically significant increase in the amount of γ-H2AX foci (1.5 ± 1.6 foci/nucleus in SiR-DNA–treated cells without SCBs, compared with 0.7 ± 1.0 foci/nucleus in control cells) (Fig 4D and E).

Taken together, these results demonstrate that SiR-DNA induces DNA damage by promoting non-centromeric chromosome entanglement that results in SCBs during anaphase.

## Discussion

Hoechst dyes have been previously reported to induce phototoxicity during time-lapse fluorescence microscopy (5), and alterations in DNA conformation and DNA damage when exposed to UV light (13). Moreover, Hoechst 33342-stained cell nuclei and chromatin exhibited an increased occurrence of DNA-protein (mainly core histones) crosslinks upon UV illumination (14). Despite it was reported that it can induce DNA damage and impair cell cycle progression (6), a Hoechst-based dye, SiR-DNA, has been widely recognized as a valuable tool that enabled multi-channel live-cell imaging of various cellular processes.

In this study, we examined the effect of SiR-DNA–based live-cell imaging on chromosome segregation and cell fate of dividing cells. We show that SiR-DNA induces a dose- and time-dependent non-centromeric chromosome entanglement that results in formation of SCBs during anaphase of four studied human cell lines. Because of the generation of SCBs, SiR-DNA staining interfered with chromosome segregation, and, consequently, impaired anaphase spindle elongation. The unresolved SCBs resulted in DNA damage that was passed forward to the following cell cycle, thereby having a detrimental effect on genome integrity. Importantly, we show that the formation of SiR-DNA–induced SCBs depends on the imaging-based light irradiation and is consequently sensitive to the time-intervals used during imaging, as the phenotype was minimized using shorter imaging frequencies. This may explain why SiR-DNA was initially reported to have minimal cytotoxicity upon far-red light illumination and why no chromosome segregation problems were detected when cells were imaged using five (1) or eight (6) minutes time-intervals. However, detailed studying of mitosis-related process often requires time-lapse imaging using a time interval shorter than 5 min, which, as evident from the results presented in this study, can severely interfere with chromosome segregation and genome integrity.

Our study shows that the incidence of SCBs can be reduced by using the lowest possible SiR-DNA concentration and/or shorter imaging frequencies. Alternatively, the light irradiation can be reduced by imaging with lower magnification, lower laser power, and lower exposure time, and by selecting less z-planes, all of which also affect the quality of imaging, via its effect on resolution and/or signal-to-noise ratio. Thus, although it is possible to lower the dye concentration and light irradiation to reduce the negative effects of SiR-DNA–based live-cell imaging, one has to be very careful when designing the experiments. Although we have not noticed significant SiR-DNA–induced problems before anaphase, suggesting that SiR-DNA is a valuable tool for live-cell imaging of early mitosis, the potential risks and limitations of this fluorescent probe should be carefully considered when designing experiments and interpreting the results.

In conclusion, our findings highlight the drawbacks in using SiR-DNA for investigation of late mitotic events and DNA damage-related topics and urge the use of alternative labeling strategies to study these processes.

## Materials and Methods

### Cell culture

Human osteosarcoma cell lines U2OS (Danish Cancer Institute cell line bank), U2OS cells stably expressing H2B-GFP/mCherry-α-tubulin (15), U2OS cells stably expressing H2B-GFP (gift from S Geley, Innsbruck Medical University, Austria), U2OS cells stably expressing CENP-A-GFP/mCherry-α-tubulin (15), colorectal adenocarcinoma cell line DLD-1 (Danish Cancer Institute cell line bank), and HeLa Kyoto (Danish Cancer Institute cell line bank) cells were grown in DMEM (Gibco). The immortalized hTERT-RPE-1 (RPE-1) cells (ATCC) and RPE-1 H2B-GFP/RFP-α-tubulin cell lines (gift from H

**Table 1.** Summary of the quantifications, number of cells, and experiments.

| Fig 1B | | | | |
|---|---|---|---|---|
| **Cell line** | **Condition** | **Rate of severe chromatin bridges (mean ± SD) (%)** | **Rate of chromatin bridges (mean ± SD) (%)** | **No. of cells (No. of experiments)** |
| RPE1 | AdV-H2B | 0.00 ± 0.00 | 3.73 ± 6.41 | 34 (3) |
| | 20 nM SiR-DNA 2 h | 16.04 ± 9.90 | 4.79 ± 4.18 | 42 (3) |
| | 20 nM SiR-DNA ON | 10.85 ± 10.72 | 6.08 ± 5.63 | 46 (3) |
| | 100 nM SiR-DNA 30 min | 36.31 ± 28.93 | 3.84 ± 3.36 | 52 (3) |
| | 100 nM SiR-DNA 2 h | 58.89 ± 18.36 | 4.45 ± 3.85 | 50 (3) |
| | 100 nM SiR-DNA ON | 37.79 ± 13.78 | 4.25 ± 4.02 | 52 (3) |
| | 500 nM SiR-DNA 30 min | 57.94 ± 8.36 | 3.52 ± 3.06 | 52 (3) |
| | 500 nM SiR-DNA 2 h | 73.28 ± 13.57 | 6.08 ± 5.63 | 29 (3) |
| | 500 nM SiR-DNA ON | 61.01 ± 12.12 | 2.78 ± 4.81 | 49 (3) |
| U2OS | AdV-H2B | 0.00 ± 0.00 | 22.81 ± 5.97 | 49 (3) |
| | 20 nM SiR-DNA 2 h | 16.87 ± 2.76 | 21.29 ± 9.78 | 52 (3) |
| | 20 nM SiR-DNA ON | 7.51 ± 7.70 | 18.61 ± 7.09 | 78 (3) |
| | 100 nM SiR-DNA 30 min | 34.47 ± 8.34 | 27.02 ± 5.58 | 40 (3) |
| | 100 nM SiR-DNA 2 h | 45.38 ± 5.04 | 22.51 ± 7.98 | 61 (3) |
| | 100 nM SiR-DNA ON | 17.87 ± 1.08 | 32.31 ± 11.90 | 72 (3) |
| | 500 nM SiR-DNA 30 min | 87.88 ± 10.50 | 6.06 ± 5.25 | 26 (3) |
| | 500 nM SiR-DNA 2 h | 88.43 ± 11.14 | 4.17 ± 7.22 | 26 (3) |
| | 500 nM SiR-DNA ON | 53.16 ± 16.71 | 16.24 ± 16.16 | 61 (3) |
| DLD1 | AdV-H2B | 0.00 ± 0.00 | 10.72 ± 4.10 | 48 (3) |
| | 20 nM SiR-DNA 2 h | 21.28 ± 12.12 | 8.44 ± 9.16 | 40 (3) |
| | 20 nM SiR-DNA ON | 15.00 ± 6.01 | 5.55 ± 4.81 | 29 (3) |
| | 100 nM SiR-DNA 30 min | 20.00 ± 10.00 | 10.00 ± 10.00 | 30 (3) |
| | 100 nM SiR-DNA 2 h | 41.98 ± 10.63 | 23.88 ± 7.99 | 39 (3) |
| | 100 nM SiR-DNA ON | 16.99 ± 5.58 | 3.81 ± 3.31 | 53 (3) |
| | 500 nM SiR-DNA 30 min | 54.55 ± 7.88 | 19.95 ± 4.44 | 35 (3) |
| | 500 nM SiR-DNA 2 h | 80.08 ± 3.21 | 17.35 ± 2.38 | 35 (3) |
| | 500 nM SiR-DNA ON | 18.95 ± 2.78 | 15.53 ± 0.46 | 58 (3) |
| HeLa | AdV-H2B | 0.00 ± 0.00 | 9.11 ± 1.99 | 34 (3) |
| | 20 nM SiR-DNA 2 h | 15.11 ± 4.41 | 17.94 ± 1.80 | 81 (3) |
| | 20 nM SiR-DNA ON | 0.00 ± 0.00 | 5.97 ± 5.17 | 63 (3) |
| | 100 nM SiR-DNA 30 min | 67.10 ± 5.78 | 2.56 ± 4.44 | 45 (3) |
| | 100 nM SiR-DNA 2 h | 72.48 ± 3.29 | 7.26 ± 2.90 | 71 (3) |
| | 100 nM SiR-DNA ON | 31.39 ± 9.87 | 10.33 ± 5.51 | 119 (3) |
| | 500 nM SiR-DNA 30 min | 83.05 ± 5.99 | 0.00 ± 0.00 | 28 (3) |
| | 500 nM SiR-DNA 2 h | 77.03 ± 10.42 | 3.94 ± 3.43 | 47 (3) |
| | 500 nM SiR-DNA ON | 41.82 ± 23.09 | 7.16 ± 8.28 | 76 (3) |

Maiato, Institute for Research and Innovation in Health–i3S, Porto, Portugal) were grown in DMEM/F12 (Gibco). In all cases, the medium was supplemented with 10% FBS, (Invitrogen) and cells were maintained at 37°C in an atmosphere of 5% $CO_2$.

## Live-cell imaging

To characterize the effects of SiR-DNA during mitosis, spinning-disk confocal live-cell time-lapse imaging was performed using a Plan-

**Table 1.   Continued**

| Fig 2B | | | | |
|---|---|---|---|---|
| **Cell line** | **Condition** | **Rate of severe chromatin bridges (mean ± SD) (%)** | **Rate of chromatin bridges (mean ± SD) (%)** | **No. of cells (No. of experiments)** |
| RPE1 H2B-GFP/RFP-α-Tub | CTRL + 640 nm | 0.00 ± 0.00 | 0.00 ± 0.00 | 46 (3) |
| | SiR-DNA | 0.00 ± 0.00 | 1.52 ± 2.63 | 62 (3) |
| | SiR-DNA + 640 nm | 48.15 ± 3.21 | 0.00 ± 0.00 | 39 (3) |
| | SiR-DNA + 640 nm (5 min) | 12.34 ± 2.65 | 2.56 ± 4.44 | 68 (3) |
| U2OS H2B-GFP/mCh-α-Tub | CTRL + 640 nm | 0.00 ± 0.00 | 25.00 ± 8.33 | 42 (3) |
| | SiR-DNA | 0.00 ± 0.00 | 28.62 ± 9.34 | 46 (3) |
| | SiR-DNA + 640 nm | 66.07 ± 4.26 | 19.69 ± 4.82 | 39 (3) |
| | SiR-DNA + 640 nm (5 min) | 14.49 ± 2.09 | 23.63 ± 7.44 | 48 (3) |

| Fig 3C and D | | | |
|---|---|---|---|
| **Cell line** | **Condition** | **Pole-pole velocity (mean ± SD) (µm/min)** | **No. of cells (No. of experiments)** |
| RPE1 H2B-GFP/RFP-α-Tub | CTRL | 1.64 ± 0.41 | 45 (3) |
| | 100 nM SiR-DNA w/o SCB | 1.51 ± 0.19 | 19 (3) |
| | 100 nM SiR-DNA with SCB | 0.44 ± 0.34 | 19 (3) |
| U2OS H2B-GFP/mCh-α-Tub | CTRL | 1.23 ± 0.29 | 27 (3) |
| | 100 nM SiR-DNA w/o SCB | 1.15 ± 0.25 | 25 (3) |
| | 100 nM SiR-DNA with SCB | 0.23 ± 0.19 | 26 (3) |

| Fig 4B and E | | | |
|---|---|---|---|
| **Cell line** | **Condition** | **Rate of SCB w/o CENP-A (mean ± SD) (%)** | **Rate of SCB with CENP-A (mean ± SD) (%)** | **No. of cells (no. of experiments)** |
| U2OS CENP-A-GFP/mCh-α-Tub | 100 nM SiR-DNA 2 h | 74.92 ± 19.95 | 25.08 ± 19.95 | 79 (5) |
| **Cell line** | **Condition** | **Number of γ-H2AX foci per nucleus (mean ± SD)** | | **No. of cells (no. of experiments)** |
| U2OS H2B-GFP | CTRL | 0.7 ± 1.0 | | 46 (3) |
| | 500 nM SIR-DNA w/o SCB | 1.5 ± 1.6 | | 35 (3) |
| | 500 nM SIR-DNA with SCB | 4.0 ± 3.5 | | 36 (3) |

| Fig S1B and C | | | |
|---|---|---|---|
| **Cell line** | **Condition** | **Signal-to-noise ratio (mean ± SD)** | **No. of cells (no. of experiments)** |
| RPE1 | 20 nM SiR-DNA 2 h | 1.15 ± 0.02 | 90 (3) |
| | 20 nM SiR-DNA ON | 1.16 ± 0.05 | 90 (3) |
| | 100 nM SiR-DNA 30 min | 1.26 ± 0.28 | 90 (3) |
| | 100 nM SiR-DNA 2 h | 1.68 ± 0.21 | 90 (3) |
| | 100 nM SiR-DNA ON | 1.18 ± 0.06 | 90 (3) |
| | 500 nM SiR-DNA 30 min | 1.60 ± 0.34 | 90 (3) |
| | 500 nM SiR-DNA 2 h | 2.02 ± 0.29 | 90 (3) |
| | 500 nM SiR-DNA ON | 1.64 ± 0.15 | 90 (3) |

**Table 1.   Continued**

| Cell line | Condition | | No. of cells (no. of experiments) |
|---|---|---|---|
| U2OS | 20 nM SiR-DNA 2 h | 1.12 ± 0.01 | 90 (3) |
| | 20 nM SiR-DNA ON | 1.09 ± 0.06 | 90 (3) |
| | 100 nM SiR-DNA 30 min | 1.37 ± 0.10 | 90 (3) |
| | 100 nM SiR-DNA 2 h | 1.56 ± 0.20 | 90 (3) |
| | 100 nM SiR-DNA ON | 1.33 ± 0.14 | 90 (3) |
| | 500 nM SiR-DNA 30 min | 1.71 ± 0.03 | 90 (3) |
| | 500 nM SiR-DNA 2 h | 1.90 ± 0.08 | 90 (3) |
| | 500 nM SiR-DNA ON | 1.34 ± 0.05 | 90 (3) |
| DLD1 | 20 nM SiR-DNA 2 h | 1.12 ± 0.07 | 90 (3) |
| | 20 nM SiR-DNA ON | 1.09 ± 0.03 | 90 (3) |
| | 100 nM SiR-DNA 30 min | 1.12 ± 0.03 | 90 (3) |
| | 100 nM SiR-DNA 2 h | 1.39 ± 0.08 | 90 (3) |
| | 100 nM SiR-DNA ON | 1.11 ± 0.05 | 90 (3) |
| | 500 nM SiR-DNA 30 min | 1.57 ± 0.14 | 90 (3) |
| | 500 nM SiR-DNA 2 h | 1.64 ± 0.11 | 90 (3) |
| | 500 nM SiR-DNA ON | 1.33 ± 0.05 | 90 (3) |
| HeLa | 20 nM SiR-DNA 2 h | 1.18 ± 0.06 | 90 (3) |
| | 20 nM SiR-DNA ON | 1.04 ± 0.02 | 90 (3) |
| | 100 nM SiR-DNA 30 min | 1.67 ± 0.16 | 90 (3) |
| | 100 nM SiR-DNA 2 h | 2.33 ± 0.34 | 90 (3) |
| | 100 nM SiR-DNA ON | 1.21 ± 0.11 | 90 (3) |
| | 500 nM SiR-DNA 30 min | 3.60 ± 0.39 | 90 (3) |
| | 500 nM SiR-DNA 2 h | 3.88 ± 0.37 | 90 (3) |
| | 500 nM SiR-DNA ON | 1.67 ± 0.10 | 90 (3) |

| Cell line | Condition | NEB to AO duration (mean ± SD) (min) | No. of cells (no. of experiments) |
|---|---|---|---|
| RPE1 | AdV-H2B | 20 ± 2 | 26 (3) |
| | 20 nM SiR-DNA 2 h | 21 ± 4 | 23 (3) |
| | 20 nM SiR-DNA ON | 23 ± 6 | 30 (3) |
| | 100 nM SiR-DNA 30 min | 21 ± 6 | 35 (3) |
| | 100 nM SiR-DNA 2 h | 24 ± 5 | 32 (3) |
| | 100 nM SiR-DNA ON | 24 ± 8 | 44 (3) |
| | 500 nM SiR-DNA 30 min | 24 ± 8 | 30 (3) |
| | 500 nM SiR-DNA 2 h | 29 ± 10 | 13 (3) |
| | 500 nM SiR-DNA ON | 26 ± 7 | 25 (3) |

| | | | |
|---|---|---|---|
| | AdV-H2B | 33 ± 6 | 21 (3) |
| | 20 nM SiR-DNA 2 h | 27 ± 8 | 22 (3) |
| | 20 nM SiR-DNA ON | 33 ± 7 | 26 (3) |
| | 100 nM SiR-DNA 30 min | 23 ± 6 | 21 (3) |
| U2OS | 100 nM SiR-DNA 2 h | 26 ± 6 | 22 (3) |
| | 100 nM SiR-DNA ON | 33 ± 13 | 21 (3) |
| | 500 nM SiR-DNA 30 min | 36 ± 5 | 13 (3) |
| | 500 nM SiR-DNA 2 h | 31 ± 9 | 14 (3) |
| | 500 nM SiR-DNA ON | 32 ± 6 | 18 (3) |
| | AdV-H2B | 28 ± 7 | 21 (3) |
| | 20 nM SiR-DNA 2 h | 29 ± 7 | 18 (3) |
| | 20 nM SiR-DNA ON | 34 ± 9 | 19 (3) |
| | 100 nM SiR-DNA 30 min | 31 ± 7 | 20 (3) |
| DLD1 | 100 nM SiR-DNA 2 h | 35 ± 8 | 21 (3) |
| | 100 nM SiR-DNA ON | 33 ± 6 | 21 (3) |
| | 500 nM SiR-DNA 30 min | 35 ± 7 | 22 (3) |
| | 500 nM SiR-DNA 2 h | 33 ± 8 | 20 (3) |
| | 500 nM SiR-DNA ON | 32 ± 7 | 23 (3) |
| | AdV-H2B | 35 ± 6 | 28 (3) |
| | 20 nM SiR-DNA 2 h | 28 ± 7 | 21 (3) |
| | 20 nM SiR-DNA ON | 32 ± 7 | 18 (3) |
| | 100 nM SiR-DNA 30 min | 35 ± 10 | 20 (3) |
| HeLa | 100 nM SiR-DNA 2 h | 39 ± 14 | 30 (3) |
| | 100 nM SiR-DNA ON | 35 ± 10 | 36 (3) |
| | 500 nM SiR-DNA 30 min | 42 ± 9 | 21 (3) |
| | 500 nM SiR-DNA 2 h | 43 ± 13 | 15 (3) |
| | 500 nM SiR-DNA ON | 42 ± 11 | 15 (3) |

Apochromat DIC 63x/1.4 NA oil objective mounted on an inverted Zeiss Axio Observer Z1 microscope (Marianas Imaging Workstation from 3i-Intelligent Imaging and Innovations Inc.), equipped with a CSU-X1 spinning-disk confocal head (Yokogawa Corporation of America) and four laser lines (405, 488, 561, and 640 nm) in an environment-controlled chamber (37°C with controlled humidity and 5% $CO_2$ supply). Images were acquired using an iXon Ultra 888 EM-CCD camera (Andor Technology) as described before ([16]). Briefly, RPE-1, U2OS, DLD-1, and HeLa cells were cultured in 96-well glass-bottomed plates (P96-1.5H-N; Cellvis) and treated with differing concentrations of SiR-DNA (Spirochrome) for either 30 min, 2 h or overnight (ON) (14–18 h) before imaging. For experiments involving U2OS and DLD-1 cell lines, 10 $\mu$M verapamil (Spirochrome) was added to block efflux pumps.

Adenovirus expressing H2B-RFP (AdV-H2B-RFP) used to enable DNA visualization in control conditions were produced according to the manufacturer's instructions (pAd/CMV/V5-DEST Gateway Vector Kit; Thermo Fisher Scientific) ([17]). For live-cell imaging, the target cells were infected with the purified adenovirus particles for 4–6 h, washed thoroughly with fresh DMEM,

and incubated overnight. Cells were imaged at a 1 min interval for 90–120 min with 1-$\mu$m z-plane slices, covering the entire mitotic spindle.

To study light irradiation dependency, RPE-1 H2B-GFP/RFP-$\alpha$-tubulin and U2OS H2B-GFP/mCherry-$\alpha$-tubulin cells were imaged in the presence or absence of 100 nM SiR-DNA after 2 h incubation, either with or without activation of the 640 nm laser at a 2- or 5-min interval.

To evaluate anaphase spindle elongation, RPE-1 H2B-GFP/RFP-$\alpha$-tubulin and U2OS H2B-GFP/mCherry-$\alpha$-tubulin cells were imaged in the presence or absence of 100 nM SiR-DNA after 2 h of incubation at a 2-min interval. To monitor centromere separation during anaphase, U2OS CENP-A-GFP/mCherry-$\alpha$-tubulin cells were imaged as described above.

## Correlative imaging

For DNA damage evaluation, we measured the induction of the DNA damage marker γ-H2AX (H2AX pS139) using correlative microscopy. To this end, U2OS H2B-GFP cells were seeded on a

glass-bottom 96-well plate (P96-1.5H-N; Cellvis) and arrested in the G2 phase using 5 $\mu$M RO-3306 (Sigma-Aldrich) with overnight incubation. Next day, the arrested cells were treated with 500 nM SiR-DNA for 30 min followed by release into media containing 500 nM SiR-DNA. Synchronized cells were imaged for 2 h at 2-min interval using a ZEISS Celldiscoverer 7 microscope (Carl Zeiss) in an environment-controlled chamber (37°C with controlled humidity and 5% $CO_2$ supply), using a Plan-Apochromat 20x/0.95 NA autocorrected objective. Five 2-$\mu$m-separated z-planes of mitotic cells were acquired. After live-cell imaging, cells were fixed with 4% PFA (VWR) for 20 min at RT and immunostained as described before ([18]). Briefly, the cells were blocked with PBS containing 0.5% Tween-20 and 1% FBS for 30 min (PBS-TF). Cells were then incubated with $\gamma$-H2AX primary antibody (anti-$\gamma$-H2AX, pS139, rabbit polyclonal; Cell Signaling) diluted in PBS-TF at RT for 1 h and washed with PBS for 5 min. Cells were incubated with anti-mouse Alexa Fluor 568 secondary antibody diluted in PBS-TF for 30 min at RT. DAPI (Sigma-Aldrich) was used DNA was counterstained at 1 $\mu$g/ml concentration. Immunostained cells were imaged using the ZEISS Celldiscoverer 7 microscope. Postmitotic cells were traced using saved co-ordinates and the number of $\gamma$-H2AX foci per nucleus was quantified manually using ZEN 3.0 blue edition software (Carl Zeiss).

### Quantification and statistical analysis

Mitotic duration was quantified by tracking the time spent from nuclear envelope breakdown to anaphase onset.

The mean intensity of SiR-DNA was quantified by drawing an ROI around the interphase nucleus and the extracellular mean intensity was extracted from the ROI using ImageJ (National Institute of Health, Bethesda, MD, USA). The signal-to-noise ratio was quantified as the ratio between the two mean intensities.

The pole–pole distance was quantified by measuring the 3D distance between the two poles from times-lapse images of metaphase spindle and through the progression of anaphase as described before ([19]) to extract the $\Delta$ pole–pole distance. The pole–pole velocity was calculated from the net difference in spindle length ($\Delta$d) from metaphase to 2 min after anaphase onset using the formula $\Delta d/t$ where t is time in minutes. GraphPad Prism 9.0.0 was used to generate statistical analysis and graphs. Figure legends provide information on the statistical details of the experiments and tests used. The normality of the data points was checked using the Shapiro–Wilk test. $t$ test (unpaired, two-tailed; normal distribution) or Mann–Whitney $U$ test (unpaired, two-tailed; no normal distribution) was used to determine statistical significance. $F$ test was used to compare variances, and Welch's correction was applied when variances were unequal. Figures and figure legends contain details on statistical significance and n values for each condition are listed in Table 1.

## Data Availability

This study includes no data deposited in external repositories.

## Supplementary Information

## Acknowledgements

We thank Stephan Geley and Helder Maiato for reagents. We thank Martina Barisic for excellent technical assistance. We thank the Bioimaging Core Facility at the Danish Cancer Institute for support in fluorescence imaging. Work in the laboratory of M Barisic is supported by grants from the Lundbeck Foundation (R215-2015-4081) and the Novo Nordisk Foundation (NNF19OC0058504).

### Author Contributions

G Rajendraprasad: investigation, visualization, methodology, and writing—original draft, review, and editing.
S Rodriguez-Calado: investigation, visualization, methodology, and writing—original draft, review, and editing.
M Barisic: conceptualization, supervision, funding acquisition, methodology, and writing—original draft, review, and editing.

### Conflict of Interest Statement

The authors declare that they have no conflict of interest.

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
