## [Reviewer comments · Life Science Alliance]

Life Science Alliance

SiR-DNA/SiR-Hoechst-induced chromosome entanglement generates severe anaphase bridges and DNA damage

Girish Rajendraprasad, Sergi Rodriguez-Calado, and Marin Barisic

DOI: <https://doi.org/10.26508/lsa.202302260>

Corresponding author(s): Marin Barisic, Danish Cancer Institute and Girish Rajendraprasad, Danish Cancer Society Research Center

Review Timeline:

Submission Date:	2023-07-08
Editorial Decision:	2023-08-21
Revision Received:	2023-09-01
Editorial Decision:	2023-09-06
Revision Received:	2023-09-07
Accepted:	2023-09-07

Transaction Report:

August 21, 2023

Re: Life Science Alliance manuscript #LSA-2023-02260-T

Marin Barisic
Danish Cancer Society Research Center
Cell Division and Cytoskeleton
Strandboulevarden 49
Copenhagen, Denmark

Dear Dr. Barisic,

Thank you for submitting your manuscript entitled "SiR-DNA/SiR-Hoechst-induced chromosome entanglement generates severe anaphase bridges that cause DNA damage" to Life Science Alliance. The manuscript was assessed by expert reviewers, whose comments are appended to this letter. We invite you to submit a revised manuscript addressing the Reviewer comments.

Thank you for this interesting contribution to Life Science Alliance. We are looking forward to receiving your revised manuscript.

Sincerely,

B. MANUSCRIPT ORGANIZATION AND FORMATTING:

Reviewer #1 (Comments to the Authors (Required)):

In this very well written and clear manuscript, Rajendraprasad and colleagues highlight the need for revisiting the risks and limitations of using SiR-DNA when studying late mitotic processes.

In brief, using live-cell imaging and immunofluorescence analysis on a panel of commonly used human cell lines, the authors show that SiR-DNA-based chromosome labeling produces strong side-effects on chromosome segregation. The authors propose that, light-irradiated SiR-DNA promotes non-centromeric chromosome entanglement, which leads to chromatin bridges in anaphase that consequently hinder anaphase spindle elongation and genome integrity.

The experiments were carefully controlled and the conclusions, for the most part, are well supported by the data. Given the widespread use of SiR-DNA for live-cell imaging of interphase chromatin and mitotic chromosomes, this manuscript will most likely represent an important piece of cautionary information to the research community.

I just have two comments that the authors may wish to consider:

1-Please make it clear in the text as well as in Figure 4E and in Table EV1. that the number of γ H2AX foci determined, are actually the number of γ H2AX foci per nucleus.

2-There is a typo in the abstract "In oarticular, we found that..."

Reviewer #2 (Comments to the Authors (Required)):

Fluorescent dyes are useful for live cell imaging, but their cytotoxicity through binding to cellular structures and exposure to light must be carefully considered. SiR-DNA, a cell-permeable DNA labelling probe excited by far-red light, has lower cytotoxicity compared to UV-excited Hoechst 33342 and is now widely used. In this manuscript, Rajendraprasad et al. investigated the effects of SiR-DNA on mitosis in human cell lines. They found dose-, time- and light-dependent chromosome entanglement that impaired chromosome segregation and spindle elongation in anaphase, resulting in DNA damage, highlighting a caveat in the use of SiR-DNA.

The results reported in the manuscript provide useful information for researchers working on mitosis using SiR-DNA. The experiments were well designed and performed to a high standard. However, as a research topic, it is difficult to consider the manuscript as more than a pilot experiment for live cell imaging using SiR-DNA. When using fluorescent dyes, we must always be aware of the possibility of cytotoxicity and try to avoid it as much as possible. The authors' suggestion to reduce the concentration of SiR-DNA or light exposure is standard practice in live cell imaging. It might be worth reporting if our current understanding of the mitotic process were distorted by the cytotoxicity of SiR-DNA, but the chromosome entanglement phenotype shown here is obviously abnormal and would not be misinterpreted by researchers. It may be interesting to elucidate the mechanism of chromosome entanglement by SiR-DNA by considering the interaction mode of DNA and SiR-DNA and comparing it with Hoechst 33342. In terms of phototoxicity, comparison with recently developed fluorescent dyes such as SPY650-DNA and SiR700-DNA will also be informative, but its scientific significance is still doubtful. Although the topic of the manuscript will attract the attention of researchers in the field of mitosis, I am afraid that its findings are not sufficient for a full paper.

Reviewer #1:

In this very well written and clear manuscript, Rajendraprasad and colleagues highlight the need for revisiting the risks and limitations of using SiR-DNA when studying late mitotic processes. In brief, using live-cell imaging and immunofluorescence analysis on a panel of commonly used human cell lines, the authors show that SiR-DNA-based chromosome labeling produces strong side-effects on chromosome segregation. The authors propose that, light-irradiated SiR-DNA promotes non-centromeric chromosome entanglement, which leads to chromatin bridges in anaphase that consequently hinder anaphase spindle elongation and genome integrity.

The experiments were carefully controlled and the conclusions, for the most part, are well supported by the data. Given the widespread use of SiR-DNA for live-cell imaging of interphase chromatin and mitotic chromosomes, this manuscript will most likely represent an important piece of cautionary information to the research community.

We thank the reviewer for recognizing the quality of our experiments and our manuscript, as well as its potential to represent an important piece of cautionary information to the research community.

I just have two comments that the authors may wish to consider:

1-Please make it clear in the text as well as in Figure 4E and in Table EV1. that the number of γ H2AX foci determined, are actually the number of γ H2AX foci per nucleus.

We thank the reviewer for pointing to this omission, which we have corrected in the revised version of our manuscript.

2-There is a typo in the abstract "In oarticular, we found that..."

We have corrected this in the revised manuscript.

Reviewer #2:

Fluorescent dyes are useful for live cell imaging, but their cytotoxicity through binding to cellular structures and exposure to light must be carefully considered. SiR-DNA, a cell-permeable DNA labelling probe excited by far-red light, has lower cytotoxicity compared to UV-excited Hoechst 33342 and is now widely used. In this manuscript, Rajendraprasad et al. investigated the effects of SiR-DNA on mitosis in human cell lines. They found dose-, time- and light-dependent chromosome entanglement that impaired chromosome segregation and spindle elongation in anaphase, resulting in DNA damage, highlighting a caveat in the use of SiR-DNA.

The results reported in the manuscript provide useful information for researchers working on mitosis using SiR-DNA. The experiments were well designed and performed to a high standard.

We thank the reviewer for recognizing the high quality of our experiments and that our study provides useful information for researchers working on mitosis.

However, as a research topic, it is difficult to consider the manuscript as more than a pilot experiment for live cell imaging using SiR-DNA. When using fluorescent dyes, we must always be aware of the possibility of cytotoxicity and try to avoid it as much as possible. The authors' suggestion to reduce the concentration of SiR-DNA or light exposure is standard practice in live cell imaging. It might be worth reporting if our current understanding of the mitotic process were

distorted by the cytotoxicity of SiR-DNA, but the chromosome entanglement phenotype shown here is obviously abnormal and would not be misinterpreted by researchers.

We agree with the reviewer that reducing the concentration of SiR-DNA or light exposure is standard practice in live cell imaging. However, we show that if we want to follow mitosis in detail, even a very low concentration of the dye combined with a very sensitive imaging system (spinning-disk microscope equipped with an EM-CCD camera) is not enough to fully avoid the problems. Although more than 300 research articles have used SiR-DNA so far (many of them in the context of studying chromosome segregation and DNA damage), the mitotic problems described in detail in our study have been completely overlooked. Because the SiR-DNA-induced problems do not appear in 100% of the cells, we believe that there is a possibility that the toxicity of fluorescent dyes has not been always fully considered, and that the chromosome entanglement phenotype shown here may be misinterpreted by researchers under certain experimental conditions. For this reason, we agree with the reviewer that our current understanding of the mitotic process may have been somewhat distorted by the cytotoxicity of SiR-DNA. Instead of reanalyzing the previous studies, the primary aim of this study is to serve as a cautionary piece of information that may prevent such potential misinterpretations from happening in the future.

It may be interesting to elucidate the mechanism of chromosome entanglement by SiR-DNA by considering the interaction mode of DNA and SiR-DNA and comparing it with Hoechst 33342.

Although we understand the interest in understanding a more mechanistic aspect of the SiR-DNA-induced chromosome entanglement and DNA damage, we believe that this is out of scope of this manuscript, which should, as mentioned above, rather serve as a cautionary piece of information for the researchers.

In terms of phototoxicity, comparison with recently developed fluorescent dyes such as SPY650-DNA and SiR700-DNA will also be informative, but its scientific significance is still doubtful. Although the topic of the manuscript will attract the attention of researchers in the field of mitosis, I am afraid that its findings are not sufficient for a full paper.

For this study, we have chosen SiR-DNA as by far the most used Hoechst-based DNA dye, and it would require an enormous amount of time and resources to repeat presented experiments using other DNA dyes, like SPY505-DNA, SPY555-DNA, SPY595-DNA, SPY650-DNA, SPY700-DNA and SiR700-DNA.

September 6, 2023

RE: Life Science Alliance Manuscript #LSA-2023-02260-TR

Dr. Marin Barisic
Danish Cancer Institute
Cell Division and Cytoskeleton
Strandboulevarden 49
Copenhagen 2100
Denmark

Dear Dr. Barisic,

Thank you for submitting your revised manuscript entitled "SiR-DNA/SiR-Hoechst-induced chromosome entanglement generates severe anaphase bridges and DNA damage". We would be happy to publish your paper in Life Science Alliance. Please review the following points below.

A. FINAL FILES:

B. MANUSCRIPT ORGANIZATION AND FORMATTING:

****The license to publish form must be signed before your manuscript can be sent to production. A link to the electronic license to**

publish form will be sent to the corresponding author only. Please take a moment to check your funder requirements.**

Sincerely,

September 7, 2023

RE: Life Science Alliance Manuscript #LSA-2023-02260-TRR

Dr. Marin Barisic
Danish Cancer Institute
Cell Division and Cytoskeleton
Strandboulevarden 49
Copenhagen 2100
Denmark

Dear Dr. Barisic,

Thank you for submitting your Research Article entitled "SiR-DNA/SiR-Hoechst-induced chromosome entanglement generates severe anaphase bridges and DNA damage". It is a pleasure to let you know that your manuscript is now accepted for publication in Life Science Alliance. Congratulations on this interesting work.

DISTRIBUTION OF MATERIALS:

Again, congratulations on a very nice paper. I hope you found the review process to be constructive and are pleased with how the manuscript was handled editorially. We look forward to future exciting submissions from your lab.

Sincerely,
